# Metallic Nanoparticle Integrated Ternary Polymer Blend of PVA/Starch/Glycerol: A Promising Antimicrobial Food Packaging Material

**DOI:** 10.3390/polym14071379

**Published:** 2022-03-29

**Authors:** Dali Vilma Francis, Saurav Thaliyakattil, Lucy Cherian, Neeru Sood, Trupti Gokhale

**Affiliations:** Department of Biotechnology, BITS Pilani Dubai Campus, Dubai International Academic City, Dubai P.O. Box 345055, United Arab Emirates; p20170903@dubai.bits-pilani.ac.in (D.V.F.); f20170013d@alumni.bits-pilani.ac.in (S.T.); lucycherian@dubai.bits-pilani.ac.in (L.C.); sood@dubai.bits-pilani.ac.in (N.S.)

**Keywords:** CuO nanoparticles, ZnO nanoparticles, PSG, antifungal

## Abstract

Advances in food processing and food packaging play a major role in keeping food safe, increasing the shelf life, and maintaining the food supply chain. Good packaging materials that enable the safe travel of food are often non-degradable and tend to persist in the environment, thereby posing a hazard. One alternative is to synthesize biodegradable polymers with an antimicrobial property while maintaining their mechanical and thermal properties. In the present study, biodegradable composites of PVA–starch–glycerol (PSG) incorporated with CuO and ZnO nanoparticles (NPs) were prepared as PSG, PSG–Cu, PSG–Zn, and PSG–CuZn films. Scanning electron microscopy, energy dispersive x-ray analysis, and thermogravimetric analysis were performed to study and characterize these films. The water barrier properties of the films improved significantly as the hydrophobicity of the PSG–CuZn film increased by 32.9% while the water absorptivity and solubility decreased by 51.49% and 60% compared to the PSG film. The Young’s modulus of the films incorporated with CuO and ZnO nanoparticles was lower than that reported for PVA, suggesting that the film possessed higher flexibility. The thermogravimetric analysis demonstrated high thermal stability for films. Biosynthesized CuO and ZnO nanoparticles exhibited antifungal activity against vegetable and fruit spoilage fungi, and hence the fabricated polymers incorporated with nanoparticles were anticipated to demonstrate an antifungal activity. The nanoparticle incorporated films exhibited fungicidal and bactericidal activity, suggesting their role in extending the shelf life of packaged food. The result of ICP-OES studies demonstrated the steady release of ions from the polymer films, however, EDX analysis demonstrated no leaching of CuO and ZnO nanoparticles from the films, thus ruling out the possibility of nanoparticles entering the packaged food. The strawberries wrapped with the fabricated films incorporated with nanoparticles demonstrated improved shelf life and retained the nutritional quality of the fruit. Among the four films, PSG–CuZn was the most promising for food wrapping since it exhibited better water-resistance, antimicrobial, thermal, and mechanical properties.

## 1. Introduction

Post-harvest diseases in fruits and vegetables are a severe problem that account for a loss of nearly 50% of crops [1]. Most of these infections are caused by fungal plant pathogens that can be harmful to humans and animals as they produce mycotoxins. Packaging fruits and vegetables are among the most popular post-harvest practices that allocate them into unitized volumes, enabling easy handling and protecting them from the hazards of transportation and storage [2]. Packaging is hence an essential component of food processing, thereby, safeguarding food quality and sensory properties while also adding value. Food packaging materials with suitable mechanical strength, barrier characteristics, thermal stability, biodegradability, antibacterial, and antioxidant capabilities are essential for food safety, retaining the quality and extending the shelf life [3].

The packaging industries use glass, metal, paper, cardboard, and plastics as their core materials [4]. Petroleum-based plastics have been used for a long time as they are relatively inexpensive, easy to use, and have good processability and durability [5]. However, the non-biodegradability of these polymers is a serious concern as it results in polluting the environment. Biodegradable films, being eco-friendly, have replaced ecology-threatening plastics [6]. Starch, polyvinyl alcohol (PVA), polylactic acid, and polyhydroxyalkanoates are a few examples of natural polymer materials [7,8]. However, compared to petroleum-based plastic materials, biopolymer films exhibit severe drawbacks such as poor barrier qualities, mechanical properties, and processability [9]. Natural polymers are frequently combined with nanoparticles or other synthetic polymers to widen their applications [10]. Blending PVA with biodegradable polymers such as starch may increase the degradability of the recalcitrant polymer while improving the mechanical properties [11,12,13,14,15]. PVA composite films with cellulose nanowhiskers and hemp protein have also been reported to improve the mechanical properties of PVA [16,17]. Glycerol, a plasticizer, makes a perfect blend for a new composite biopolymer with PVA and starch as it exhibits a similar solubility to PVA and starch [18]. Metal nanoparticles have been shown to possess antimicrobial properties, hence adding them to the polymer will increase the antimicrobial barriers, resulting in the enhanced shelf life of the food. Studies on ZnO [19] and CuO [20] nanoparticles have revealed them to possess antimicrobial properties, thus, the addition of these in the polymer will be ideal for food packaging [21]. The present study aimed at synthesizing a biodegradable polymer blend of PVA, starch, and glycerol embedded with CuO and ZnO nanoparticles that possessed antimicrobial properties while retaining mechanical and thermal properties comparable to a petroleum-based polymer. The new composite polymer aims to revolutionize the packaging industry with its unique application in extending the shelf life of perishable commodities such as fruits, vegetables, bakery products, etc.

## 2. Materials and Methods

### 2.1. Synthesis of CuO and ZnO Nanoparticles

CuO and ZnO nanoparticles were synthesized in the laboratory using a bacterium, *Stenotrophomonas maltophilia*. The studies were carried out in 250 mL Erlenmeyer flasks, which contained 100 mL sterile Luria Bertani broth. The broth was inoculated with 1 mL of 18 h old culture of *S. maltophilia* (O.D 0.8 at 530 nm). The inoculated broth was incubated at 30 °C at 150 rpm, for 24 h to allow the culture to grow. A total of 2 mM CuSO_4_ or ZnSO_4_ solution was added to two culture flasks, the pH of the media was adjusted to 8 using 1 M NaOH, and further incubated at 40 °C for three days at 200 rpm. After the incubation, the cells were lysed using a QSONICA sonicator with an amplitude of 40% and pulse of 15 s for 10 min under ice-cold conditions, and the cell lysate was centrifuged at 4000 rpm for 5 min. The supernatant was vacuum filtered through a 0.22 µm cellulose acetate membrane, and the filtrate was dialyzed using a snakeskin dialysis membrane (10 K MWCO). The dialyzed samples were lyophilized to obtain the CuO and ZnO nanoparticles, respectively. The nanoparticles were characterized using UV–Visible spectroscopy, particle size analysis, X-ray diffraction, and scanning electron microscopy analysis.

### 2.2. Fabrication of PVA Based Nanocomposite Films

The polymer films were prepared by mixing polyvinyl alcohol (PVA—Solution A), starch (Solution B), and glycerol (Solution C) solutions. To prepare Solution A, 0.8 g of PVA was soaked in 70 mL of deionized water (DI) for 12 h and heated at 95 °C with continuous stirring at 400 rpm for 3 h. Solution B was prepared by dissolving 0.8 g of starch in 20 mL of DI water and heated at 95 °C with continuous stirring at 400 rpm for 30 min. For the preparation of Solution C, 0.8 mL of glycerol was mixed with 9.2 mL of DI water and heated at 95 °C for 10 min. Solutions A, B, and C were mixed and maintained at 95 °C with continuous stirring at 400 rpm for 30 min to prepare the PSG polymer film. PVA, starch, and glycerol used for synthesizing the polymer were of analytical grade, purchased from Sigma-Aldrich.

CuO nanoparticles (15 mg) dissolved in 100 µL DI water were added to PSG to form the PSG–Cu film, and ZnO nanoparticles (25 mg) dissolved in 100 µL DI water were added to PSG to prepare the PSG-Zn polymer film. Nanoparticles of CuO (15 mg) and ZnO (25 mg) dissolved in 100 µL DI water were added to PSG to form the PSG-CuZn polymer composite.

After mixing solutions A, B, and C, and addition of nanoparticles, all the polymer composites were maintained at 95 °C for 30 min. Each mixture was then poured on a sterile glass plate and allowed to dry in an oven at 50 °C for 24 h until a uniform film was developed. The synthesized polymer films were placed in a clean polythene bag and stored in a desiccator until further analysis. All of the experimental analysis were performed in triplicate to test the reliability and reproducibility.

### 2.3. Characterization of Polymer Films

**XRD, FTIR, SEM, and EDX analysis of CuO and ZnO nanoparticles**.

The dialyzed supernatant was lyophilized (Buchi, Lyovapor L-200) and the lyophilized samples were analyzed for X-ray diffraction on a Bruker AXS Kappa APEX II CCD X-ray diffractometer operated at 40 kV and 40 mA with Cu Kα radiation (1.54 Å). A continuous scan mode was applied with a step width 0.020°, sampling time 57.3 s, and measurement temperature of 25 °C. The scanning range of 2θ was between 3° and 80°.

The surface morphology of the polymers was investigated under scanning electron microscopy (SEM) using a JEOL JSM-7600F FEG-SEM. A qualitative analysis of the metal nanoparticles embedded in the polymer films was performed using EDX (energy-dispersive X-ray spectroscopy) of the SEM image.

Attenuated total reflectance (ATR)-FTIR analysis of the polymers was performed on a Shimadzu FTIR to analyze the organic and inorganic components.


**Hydrodynamic properties of the polymer films.**


Polymer films with a size of 2 × 2 cm^2^ were dried at 50 °C for 2 h, and the initial weight (Wo) of the polymer film was recorded. All analyses were conducted in triplicate and the average weight and standard deviation were calculated. 

**Net moisture content** (NMC) [22].

The net moisture content was calculated using the constant dry weight (Wd) obtained by drying the samples at 60 °C for 24 h.
$$\mathrm{NMC}\left(\%\right)=\frac{\mathrm{Wo}-\mathrm{Wd}}{\mathrm{Wo}}\times 100\%$$

**Water absorption capacity** (Wa) [23].

The pre-dried polymer films with weight (Wo) were immersed in 100 mL of DI water for 24 h. The films were pat dried using clean tissue paper and the weight (Wt) was recorded. The water absorption capacity of the films was calculated using the following equation.
$$\mathrm{Wa}\;\left(\%\right)=\frac{\mathrm{Wt}-\mathrm{Wo}}{\mathrm{Wo}}\times 100\%$$

**Water solubility** (Ws) [24].

The polymer films treated for the water absorption test (Wa) were dried at 60 °C for 24 h to obtain a constant weight (Wc). The following formula was used to determine the water solubility of the polymer.
$$\mathrm{Ws}\;\left(\%\right)=\frac{\mathrm{Wo}-\mathrm{Wc}}{\mathrm{Wo}}\times 100\%$$

**Water contact angle** [23].

Water contact angles of PSG, PSG–Cu, PSG–Zn, and PSG–CuZn films were determined using the Ossila contact angle goniometer. A total of 2 µL of DI water was placed on the films mounted on a flat surface of the adjustable stage using a micro-syringe. The image of the drop was captured to determine the water contact angle with a precision of 0.2 by averaging 10 readings.

**Mechanical properties** [24,25].

The mechanical properties of the polymer films such as tensile strength, Young’s modulus, and elongation break were analyzed following the ASTM standard method D-882-88 [21] using a Shimadzu Autograph AGSX-10KN. Polymer films sized 50 mm × 20 mm, with a thickness of 1 µm were used for the study. The standard formula was used to calculate the tensile strength, Young’s Modulus, and elongation break.

**Thermal stability** [26].

The thermal stability of the polymer films was analyzed using a thermogravimetric analyzer (TGA 50, Shimadzu). A total of 8 mg of the sample film was taken in a standard aluminum pan and was heated from 20 to 600 °C with a heating rate of 10 °C/min under nitrogen flow (50 cm^3^/min). The derivative curve of TGA (DTG—derivative of thermogravimetric analysis) was obtained by calculating the differential of TGA valued using a central finite difference method given below: $$\mathrm{DTG}=\frac{\left(w_{t+\Delta t}-w_{t+\Delta t}\right)}{2\Delta t}$$
where W_t+Δt_ and −W_t+Δt_ are the residual sample at time t + Δt and t − Δt, respectively; and Δt is the time interval for reading the residual sample weight.

**UV and visible light transmittance** [22].

The UV and visible light transmittance of all the sample films were measured at 254 nm (T_254_) and 600 nm (T_600_), respectively, using a Perkin Elmer UV–Visible spectrophotometer with air as a reference.

**Polymer degradation analysis** [23].

A total of 400 g autoclaved garden soil was added in multiple pots and considered for the study. The polymer films (PSG, PSG-Cu, PSG-Zn, PSG-CuZn) sized 2 cm × 2 cm were recorded for their initial weight (W_o_), placed at a depth of one inch from the surface of the potted soil, and incubated at room temperature. A total of 15 mL of water was sprayed daily to keep the soil moist. The setup was disturbed every 24 h to analyze the film. The polymer film was removed from a pot every 24 h, cleaned, dried at 50 °C for 2 h, and weighed (W_bd_).

Similar setup was maintained for unautoclaved garden soil to study the different in degradation rate if any due to microorganisms in soil. 

The study was performed for 56 days and the biodegradation rate (BDR) was calculated using the following formula.
$$\mathrm{BDR}\left(\%\right)=\frac{\mathrm{Wo}-\mathrm{Wbd}}{\mathrm{Wbo}}\times 100\%$$

**Release profile of Cu^2+^ and Zn^2+^ from the polymers** [27].

Polymer films (PSG, PSG-Cu, PSG-Zn, PSG-CuZn) sized 2 cm × 2 cm were immersed in 100 mL DI water and maintained on a shaker at 120 rpm at 30 °C. The Cu^2+^ and Zn^2+^ ions leached from the polymer were analyzed every 48 h up to 14 days using the Perkin Elmer-Optima 8300 ICP-OES. The reference solutions for Cu and Zn (1, 10, 20 ppm) were prepared from commercial ICP standards and the analysis was performed using water and 3% (*v*/*v*) nitric acid as calibration and reagent blanks, respectively. 


**Leaching of metal nanoparticles from polymer.**


Polymer films (5 cm × 5 cm) were immersed in 50 mL of DI water at 25 °C for 30 days. The water samples were filtered through a 0.22 µm nitrocellulose membrane, dialyzed using snakeskin (10 K MWCO) for 24 h against DI water, and lyophilized. EDX analysis was performed on the lyophilized samples to detect the presence of leached CuO and ZnO nanoparticles.

**Antifungal activity of CuO and ZnO Nanoparticles** [28].

*Aspergillus niger*, *Aspergillus calidoustus*, and *Penicillium chrysogenum* previously isolated in the laboratory from spoiled fruits and vegetables were considered for the study. The antifungal activity of CuO and ZnO nanoparticles was tested by spreading 100 µL of the fungal spores (OD 1.00) onto sterile potato dextrose agar and adding the copper oxide (1, 3, 5 µg) and zinc oxide (5, 7, 10 µg) nanoparticles on the surface of the plates spread with fungal spores. The plates were incubated at 30 °C for 24 h. Positive controls with 2 mM CuSO_4_ and 2 mM ZnSO_4_ and negative controls with water were also applied on the plates spread with fungal spores. The zone of inhibition developed against each fungus was measured for CuO and ZnO nanoparticles at various concentrations. 


**Food wrapping analysis.**


Strawberries were selected for the fruit wrapping analysis due to their low shelf life (5–7 days) and small size for easy wrapping [29]. Fresh strawberries purchased from the market were wrapped with PSG, PSG–Cu, PSG–Zn, and PSG–CuZn films, while a set of unwrapped strawberries was maintained as the control. Sets of 10 strawberries were maintained for each polymer and the control for statistically reliable data. All strawberries were stored at 25 °C for seven days. Observations were recorded on a daily basis by inspecting the strawberries for the spoilage with respect to the changes in surface texture. After seven days, the weight of strawberries was recorded, and the fruits were sliced and dried in a hot air oven at 40 °C for 72 h to obtain the constant dry weight. The dried samples were powdered and subjected to various biochemical and mineral content analysis.

**Estimation of total reducing sugars and total protein content** [30,31]. 

The concentration of reducing sugars and proteins was estimated by the Anthrone method and Lowrey’s method, respectively.

### 2.4. Analysis for Cu^+2^ and Zn^+2^ Ions in Strawberries

A total of 50 mg of dried strawberry sample was treated with 15 mL of 3% nitric acid and digested in a microwave-assisted digestor. The digested samples were filtered and analyzed in a Perkin Elmer-Optima 8300 ICP-OES for Cu^+2^ and Zn^+2^ ions, as previously described. 

### 2.5. Statistical Analysis

All the analyses were conducted in triplicate and the average with standard deviation was considered. The data were further evaluated using one-way analysis of variance (ANOVA) at a significance level of 0.05.

## 3. Result and Discussion

The XRD spectrum (Appendix A) confirmed the biosynthesis of CuO and ZnO nanoparticles. The diffraction peaks at 2θ values of 32.3795, 35.7643, 38.8219, 46.2562, 48.7309, 53.4758, 58.4047, 61.5645, 66.7287, and 68.191 degrees, corresponding to (111), (000), (200), (−112), (−202), (020), (202), (−113), (022), and (220) planes, confirmed the presence of copper oxide and 2θ values of 31.94, 34.64.36.42, 47.83,56.85,62.93, 68.2 (100, 002, 101, 102, 110, 103, 112) were due to the zinc oxide diffraction. The SEM images demonstrated the morphology of both CuO and ZnO nanoparticles, which were roughly spherical (inset Appendix A).

The SEM image of the PSG film (Figure 1a) exhibited a clear surface with the SEM-EDX spectra (Figure 2a), supporting the absence of any metal ions. Figure 1b–d shows the SEM images of PSG with CuO, PSG with ZnO, and PSG with CuO and ZnO nanoparticles, respectively, where the metal nanoparticles embedded in the polymer were distinctly visible, making the surface appear textured. The SEM-EDX spectra (Figure 2b–d) of the films support the presence of CuO, ZnO, and CuO–ZnO nanoparticles.

The FTIR-ATR analysis of the polymer films is given in Figure 3. FTIR studies were performed to ascertain the organic as well as inorganic components of the polymer. The peaks at 3300 cm^−1^, 2940 cm^−1^, 1720 cm^−1^, and 1030 cm^−1^ corresponded to –OH stretching, –CH sp^3^ stretching, C=O stretching, and C–O stretching, respectively [32]. These functional groups represent the organic components of the films contributed by PVA, starch, and glycerol. The peaks at 600 cm^−1^ and 1630 cm^−1^ in the PSG–Cu and PSG–CuZn films corresponded to the Cu–O vibration and stretching, respectively, while the peaks at 575 cm^−1^ and 1440 cm^−1^ in the PSG–Zn and PSG–CuZn films corresponded to the Zn–O vibration and stretching, respectively [33,34].

Most biopolymers are highly sensitive to water, and hence exposure to water may result in changes in the mechanical properties or degradation of the polymer. Hence, understanding the net moisture content, water-solubility, water absorption, and water contact angle of the developed polymer films are critical, especially when considered for food packaging applications. The net moisture content of the polymer PSG polymer film was the maximum at 17.62%, while the polymer PSG–CuZn had a content of 12.61%, which was 28.43% (±0.02) lower than PSG, indicating the potential of Cu and Zn nanoparticles to lower the ability of PSG polymer to hold free moisture. Other polymers, PSG–Cu and PSG–Zn, also exhibited a lower moisture content at 14.69% (±0.1) and 15.16% (±0.07), respectively, thereby expressing a reduced water holding capacity (Figure 4a).

The water absorption capacity (Wa) and the water solubility (Ws) of the polymers (Figure 4b) also exhibited a similar effect where the presence of CuO or ZnO nanoparticles significantly (*p* value < 0.05) lowered the ability of the polymer to absorb moisture and prevent the polymer from becoming solubilized in water. *Wa* for polymers PSG, PSG–Cu, PSG–Zn, and PSG–CuZn was observed as 30.73% 17.09%, 15.55%, and 14.9% respectively, while the *Ws* was 23.58%, 11.6%, 8.28%, and 9.3%, respectively.

PSG–CuZn exhibited a remarkable reduction in Wa and Ws by 51.49% (±0.009) and 60% (±0.017), respectively, when compared with the Wa and Ws of PSG (*p* value < 0.05). PVA, starch, and glycerol in PSG contribute hydroxyl groups, which provide free sites for water molecules to absorb via hydrogen bonding [23], resulting in an elevated Wa and Ws for the PSG polymer. Reduced Wa and Ws observed for the nanoparticle incorporated polymer could be attributed to the reduction in pore channels in the polymer films due to the incorporation of nanoparticles. The outcome of incorporating nanoparticles was the reduction in pore channels in the polymer films, thereby improving the water barrier property [35].

The water contact angle defines the hydrophobicity of the polymer. The water contact angle for PSG was observed to be the lowest while that of PSG–CuZn was the highest, signifying the former to be less hydrophobic than the latter film (Figure 4c). All films embedded with nanoparticles exhibited high hydrophobicity as the water contact angle of the films increased by 9.41%, 17.91%, and 32.9%, for PSG–Cu, PSG–Zn, and PSG–CuZn, respectively, when compared to the PSG polymer film. The surface modification of the polymer due to the incorporation of CuO and ZnO nanoparticles resulted in high water resistance, and hence increased hydrophobicity (*p* value < 0.05) of the polymer film [36].

The commercial application of any polymer depends on its mechanical properties. The four polymer films synthesized were tested for their mechanical and thermal properties to study the changes due to the incorporation of CuO and ZnO nanoparticles. Table 1 presents the mechanical properties of PSG, PSG–Cu, PSG–Zn, and PSG–CuZn films. The tensile strength of the polymers demonstrated the following order: 31.83 MPa (PSG-Zn) > 29.12 MPa (PS–-CuZn) > 23.71 MPa (PSG–Cu) > 18.05 MPa (PSG). The tensile strength of the four polymer films was higher than the reported value for PVA at 6.3 MPa [15]. Thus, the polymers embedded with CuO and ZnO nanoparticles possessed better strength than PVA. The increase in tensile strength of the polymer was attributed to the effective strain transfer to the nanoparticle–polymer interface [16].

Young’s modulus explains the rigidness of the polymer and hence a lower Young’s modulus explains the flexibility of polymer films. The polymers embedded with nanoparticles exhibited a marginally higher Young’s modulus, though not significantly different from the PSG film, as seen from the *p*-values > 0.05 for PSG–Cu (0.9842), PSG–Zn (0.613), and PSG–CuZn (0.502). The neat PVA film showed a Young’s modulus of 2.75 GPa with 57% of elongation [37]. Blending the PVA with other materials increased the elongation, thereby acting as plasticizers. Blending PVA with starch and glycerol has been reported to increase the elongation to break to 402.08%, thereby signifying the effectiveness of the components as plasticizers. The addition of the CuO and ZnO nanoparticles to PSG lowered the elongation to break marginally when compared with the PSG polymer film, which can be associated with the decrease in polymer voids due to the strong interaction between the polymer and nanoparticles [16,37,38]. The elongation to break for the PSG–Cu (*p*-value 0.0458) and PSG–Zn (*p*-value 1.2864) films was not significantly different from PSG. Thus, these results indicate improved mechanical properties in PSG polymers embedded with CuO and ZnO nanoparticles compared to PVA and PSG, thus justifying their application in the packaging industry [39].

A thermogravimetric analyzer (TGA) was used to assess the thermal stability of the polymers, and the TGA curves obtained for the four polymers are presented in Figure 4a, with the derivative thermogravimetric analysis (DTGA) curves shown in Figure 5a. The DTGA curves (Figure 5b) showed the maximum breakdown temperature (T_max_) for thermal decomposition, while the thermogravimetric curves exhibited the decomposition of the polymer films as a decrease in weight.

The polymer films were observed to undergo a multi-step thermal decomposition process. The initial thermal decomposition of the PSG polymer film was observed to be between 90 and 120 °C, which was due to water evaporation, resulting in a 0.98 mg weight loss, accounting for 12.23% total mass of the sample. The weight loss due to free moisture was lower in films embedded with nanoparticles compared to PSG. The second phase of degradation (up to about 220 °C) represents the volatilization of glycerol. The principal thermal decomposition was observed between 130 and 485 °C, which was attributed to the decomposition of starch, PVA, and glycerol molecules into a new structure through hydrogen bonding [40]. The third phase of degradation was observed from 270 to 390, with the maximum decomposition rate around 360 °C, resulting in a 6.93 mg weight loss, accounting for 81.36% total mass of the sample, followed by a fourth phase from 380–485 °C, which exhibited a small mass loss. Incorporation of the metal oxide nanoparticles insignificantly reduced the maximum degradation temperature in the third stage from 329 °C for PSG to 315 °C for PSG-CuZn (*p* value 1.16). Residual polymer after the final thermal destruction at 600 °C for the PSG, PSG-Cu, PSG-Zn, and PSG-CuZn was 4.63%, 6.74%, 8.14%, and 16.43%, respectively. It has been reported that the incorporation of nanoparticles does not significantly change the thermal decomposition pattern, but their addition to the polymer resulted in modifying the crosslinking between starch, PVA, and glycerol, which becomes more compact, thereby enhancing the thermal stability of the polymer composite films [23]. The thermogravimetric analysis of the polymer films embedded with CuO and ZnO nanoparticles did not differ significantly from the PSG film, thereby maintaining the thermal stability of the polymer.

UV light causes the oxidation of food, resulting in spoilage [22]. An ideal food packaging material should block the penetration of UV light, thereby preventing the photodegradation of food during storage and transport. Hence, it is essential to ensure that food packaging materials are barriers to UV light while transmitting visible light.

The UV transparency of PSG at 254 nm was 47.36%, which decreased further with the incorporation of CuO and ZnO nanoparticles. The polymers PSG-Cu, PSG-Zn, and PSG-CuZn exhibited a T_254_% value of 28.43%, 23.83%, and 23.09%, respectively. The polymer films incorporated with nanoparticles inevitably led to increased surface roughness, resulting in enhanced light scattering. This is advantageous as the UV barrier characteristic of the developed polymer films could better protect the food against lipid oxidation and nutrient loss [22]. The T% of the PSG film at 600 nm demonstrated a value of 93.4%, while the incorporation of nanoparticles reduced the T_600_% to 83.4%, 78.4%, and 71.7% for PSG-Cu, PSG-Zn, and PSG-CuZn, respectively. Figure 6 shows the images of the BITS Pilani logo observed through the PSG, PSG-Cu, PSG-Zn, and PSG-CuZn to understand the transparency of the polymers to visible light.

To study the rate of their degradation, the polymer films were incubated in autoclaved and non-autoclaved garden soil. Figure 7a demonstrates the rate of degradation of the polymer films incubated in sterile and non-sterile soil over 56 days, while Figure 7b shows the images of the polymer films upon degradation. The biodegradation rate of the polymer films were analyzed by studying the weight loss of the polymer films. Maximum degradation (50%) was observed for the PSG film placed in non-sterile soil, while the nanoparticle incorporated polymers exhibited a comparatively lowered degradation rate at 38–40%. This was due to the antimicrobial activity of the CuO and ZnO nanoparticles, which prevented or lowered the microbial action for degradation of the polymers, thereby increasing their stability [41]. A significant difference in the degradation rate of polymer films was observed in non-sterile soil and sterile soil, indicating the role of microorganisms in degradation. The biodegradability of biopolymers is also strongly linked to their *Wa* and *Ws*. Adsorption of water molecules onto the polymer surface encourages microorganisms such as bacteria and fungi to attach and grow on the polymer surface, followed by subsequent degradation [42]. A visible change in color from colorless to yellow, a size reduction, and a change in the texture of the films was observed on incubation. The films appeared frailer and withered due to microbial degradation after the 56^th^ day of incubation. The synthesized polymer films exhibited promising physical and mechanical properties. However, since the metal oxide nanoparticles were incorporated in some of these films, considering their application in food packaging, a study on the release of metal ions from the polymer films was essential.

Figure 8a represents the rate of Cu^+2^ and Zn^+2^ ions released from the PSG–Cu and PSG–Zn films, respectively. A total of 0.7 mg/L of Cu^+2^ ions and 0.85 mg/L of Zn^+2^ ions were released from the polymers after being submerged in water for 14 days. Similarly, the amount of Cu^+2^ and Zn^+2^ ions released from the PSG-CuZn film (Figure 8b) was comparable to those released from individual films. All polymer films exhibited a steady release of Cu^+2^ and Zn^+2^ ions, while no traces of Cu^+2^ and Zn^+2^ ions were observed from the PSG film, thus signifying the release of Cu^+2^ and Zn^+2^ ions from the incorporated CuO and ZnO nanoparticles. The recommended dietary allowance (RDA) for copper and zinc is 1400 µg and 11 µg, respectively [43,44], hence the amount of Cu^+2^ and Zn^+2^ ions released from the polymer films were much lower than the RDA by the National Institute of Health (NIH).

Cu^+2^ and Zn^+2^ ions are known to possess antimicrobial properties [45,46]. The release pattern of metal ions from the polymer when submerged in water provides great insight into the antimicrobial effects as the steady release of metal ions from the polymer can impart antimicrobial properties. The sustained release of Cu^+2^ and Zn^+2^ ions observed for all three nanoparticle incorporated/embedded polymers suggests its possible role as a packaging material with antimicrobial properties. Since the metal ions were observed to leach out from the polymer films, it would be of concern to study the leaching of nanoparticles, if any. Studying the leaching of metal nanoparticles from the polymer was important because of their inclusion in synthesized polymer films and the safety concerns of their use as food packaging material. The EDX analysis of the lyophilized water samples collected after immersing the polymer for 30 days revealed no leaching of the metal nanoparticles from the polymers as no significant peaks corresponding to CuO or ZnO were observed in the EDX spectra (S2). All four polymer films demonstrated sharp peaks for carbon and oxygen, which are associated with the polymer blend. The peaks at 0.5 and 0.8 keV for CuO and 0.5 and 1.0 keV for ZnO were distinctly missing [33,34]. Thus, the study revealed no leaching of the nanoparticles from the polymer films, suggesting them to be safe for applications in food packaging.

The antifungal effect of CuO and ZnO nanoparticles was tested against three fungal specimens *Aspergillus niger*, *Aspergillus calidoustus*, and *Penicillium chrysogenum* isolated from spoiled fruits in the laboratory. Both the nanoparticles inhibited the growth of the fungi. Figure 9 indicates the varying size of zone of inhibition (ZOI) exerted on the fungi due to the application of different concentrations of nanoparticles. The ZOI increased with the increased concentration of nanoparticles. Copper sulfate and zinc sulfate controls maintained at the same concentration as the nanoparticles did not exhibit any effect toward all the selected fungi. All three fungal specimens were found to be sensitive toward CuO nanoparticles whereas only *Aspergillus niger* was found to be resistant to ZnO nanoparticles. The minimal inhibitory concentration (MIC) and minimal fungicidal concentration (MFC) of CuO nanoparticles, ZnO nanoparticles, and Amphotericin B were 1 µg/mL, 1 µg/mL, and 2 µg/mL, respectively, against the selected fungi. The nanoparticles also possessed antibacterial activity, as checked against human pathogens *Klebsiella pneumoniae* and *Staphylococcus aureus* (Appendix A). The polymer films embedded with CuO and ZnO nanoparticles also exhibited an antifungal activity against, as anticipated (Figure 8b). PSG–Cu and PSG–CuZn exhibited a clear ZOI against *A. niger*, while PSG, PSG–Zn, and Amphotericin B (20 µg) did not inhibit the growth of *A. niger*. The nanoparticle embedded films exhibited a higher ZOI than Amphotericin B (20 µg) against *A. calidoustus*, thus signifying their application in food preservation.

Figure 10 shows the preservative effect of CuO and ZnO nanoparticles on the strawberries packaged with PSG–Cu, PSG–Zn, and PSG–CuZn polymer films. The fruits wrapped with the PSG–CuZn film appeared fresh compared to the strawberries packaged with PSG, signifying the antifungal role of nanoparticle incorporated polymer films in enhancing the shelf life of strawberries.

As the polymer demonstrated promising mechanical, thermal, and antifungal properties, this can be considered as an alternative to the petro-chemical polymers currently in use. The strawberries were also tested for the content of reducing sugar, proteins, and Cu^+2^ and Zn^+2^ ions before and after packaging to determine whether the packaging with nanoparticle incorporated polymer films had any effect on the nutritional properties of the fruits. Overall, the nutritional quality of the strawberries was better retained when packed with PSG, PSG–Cu, PSG–Zn, and PSG–CuZn when compared with the unwrapped strawberries.

The reducing sugar and protein content of strawberries lowered with increased storage time. The unwrapped strawberries exhibited the most significant reduction in content of reduced sugar 3.522 g (±0.86) and proteins 0.188 g (±0.029), whereas PSG–CuZn wrapped strawberries showed a minor reduction in reducing sugars (6.76 g (±0.54)) and protein content (0.378 (±0.038)) (Figure 11a,b). The lowered content of proteins and reducing sugars may be attributed to their breakdown due to strawberry respiration or fungal spoilage.

No significant difference was observed in the amount of Cu^+2^ and Zn^+2^ ions between strawberries wrapped with PGS or PSG with nanoparticles and unwrapped strawberries at day 0. Upon storage for seven days, the concentration of ions increased compared to day 0, which could be attributed to the moisture loss from the fruit and fungal infection (Figure 10c) [39]. Hence, the wrapping of strawberries with nanoparticle embedded polymers does not have a significant difference on the nutritional value of the fruit [47].

## 4. Conclusions

A biodegradable, antimicrobial polymer was synthesized with a blend of PVA, starch, and glycerol, embedded with CuO and ZnO nanoparticles. Addition of starch and glycerol to PVA enhanced the biodegradability and flexibility of the polymer. The polymer films characterized using SEM, EDX, and FTIR confirmed the presence of organic and inorganic components. The four polymer films exhibited improved water barrier properties with increased hydrophobicity and lowered water solubility, an important property for a packaging material. The mechanical properties of the nanoparticle embedded films exhibited increased strength and flexibility compared with the PVA and PSG films. The films demonstrated excellent thermal stability with no significant difference between the four films. Among the four blended films, PSG–CuZn exhibited improved water resistance, thermal stability, and UV barrier properties. The application of the CuO and ZnO embedded PSG in packaging was evident from the enhanced shelf life and nutritional value of strawberries wrapped in the polymer films. Studies using ICP and EDX revealed sustained release of Cu and Zn ions, but no leaching of CuO and ZnO nanoparticles. The metal ion released from the polymer embedded with nanoparticles exhibited a good antifungal property, thereby enhancing the shelf life of the strawberries. Thus, the developed polymer films embedded with nanoparticles could be considered as a promising food packaging material for ensuring the safety and increased shelf life of food, thereby meeting the global demand for food.

## Figures and Tables

**Figure 1 polymers-14-01379-f001:**
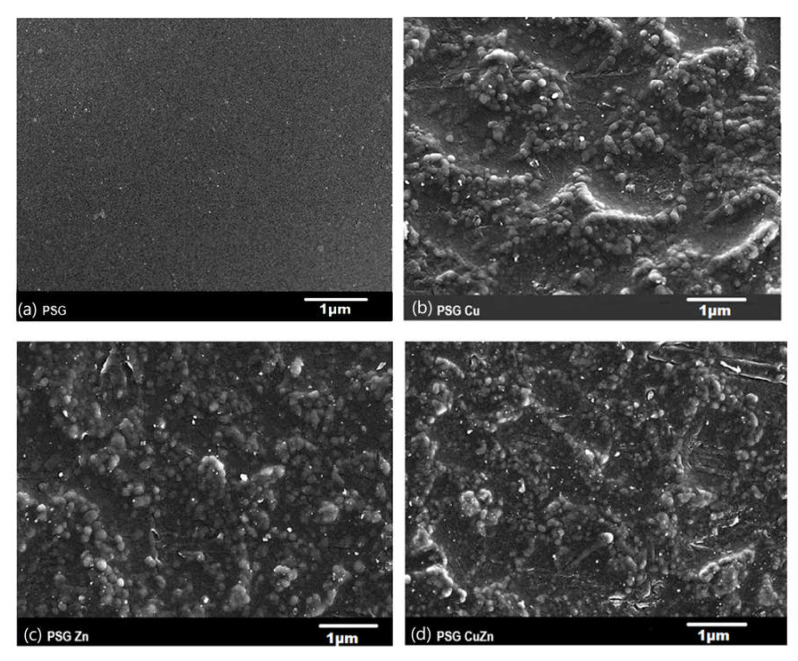
Scanning electron microscopy image of (**a**) PSG, (**b**) PSG-Cu, (**c**) PSG-Zn, and (**d**) PSG-CuZn films using a JEOL JSM-7600F FEG-SEM. All scale bars represented in 1 µm.

**Figure 2 polymers-14-01379-f002:**
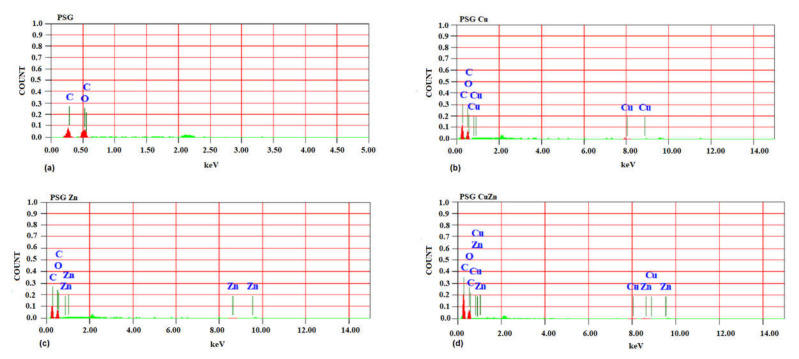
SEM-energy dispersive X-ray spectra of (**a**) PSG, (**b**) PSG-Cu, (**c**) PSG-Zn, and (**d**) PSG-CuZn films analyzing the elemental composition of developed polymer composite films.

**Figure 3 polymers-14-01379-f003:**
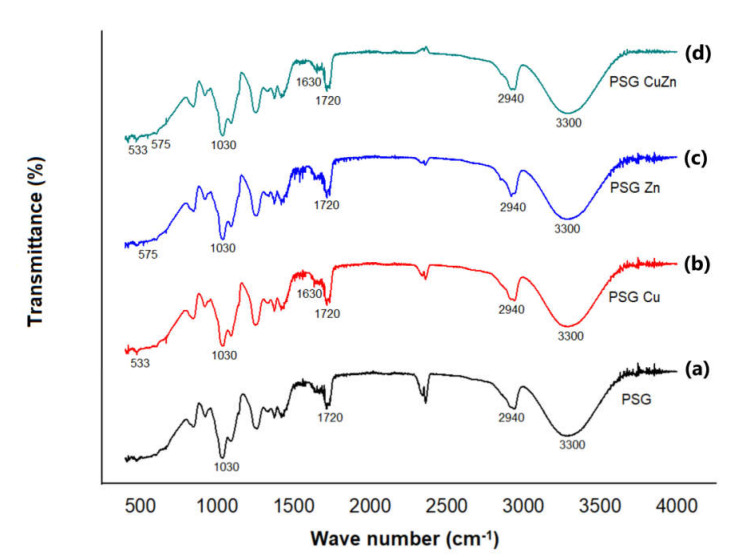
FTIR-ATR spectra of (**a**) PSG, (**b**) PSG-Cu, (**c**) PSG-Zn, and (**d**)PSG-CuZn films analyzing the organic and inorganic components of the developed polymer composite films.

**Figure 4 polymers-14-01379-f004:**
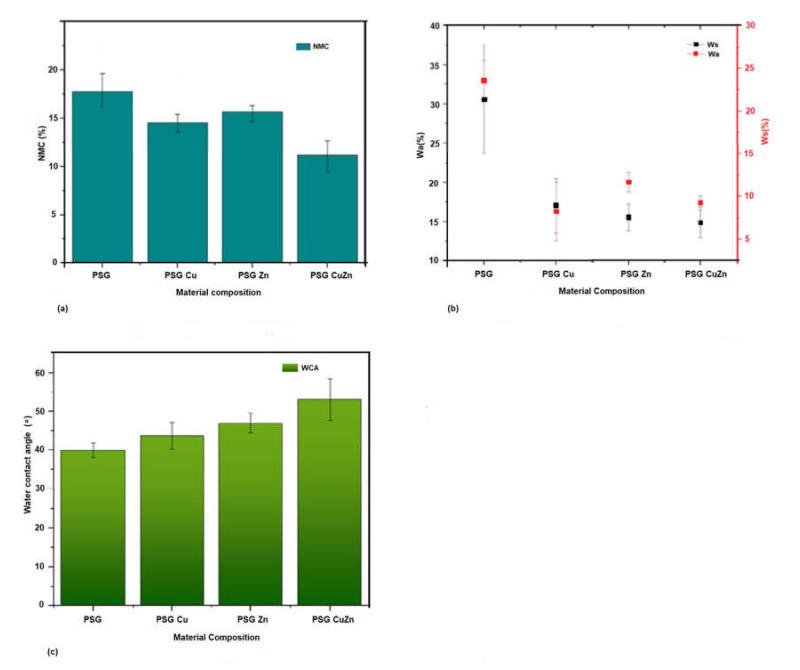
A comparison between the hydrodynamic properties of the polymer films, PSG, PSG-Cu, PSG-Zn, and PSG-CuZn for (**a**) net moisture content, (**b**) water absorbance and water solubility, (**c**) and water contact angle.

**Figure 5 polymers-14-01379-f005:**
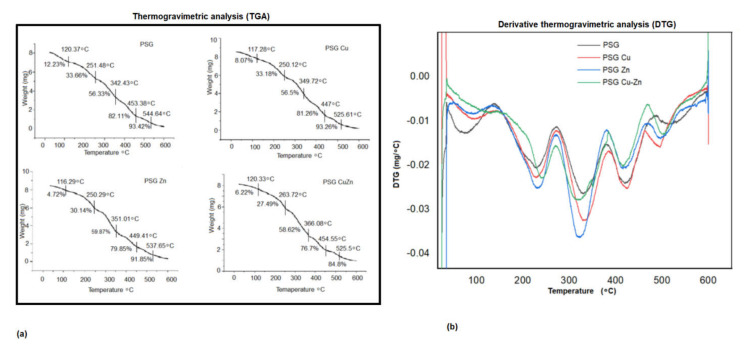
Thermogravimetric analysis of (**a**) PSG, PSG-Cu, PSG-Zn, and PSG-CuZn films. (**b**) Derivative plot of the thermogravimetric analysis of PSG, PSG–Cu, PSG–Zn, and PSG–CuZn films.

**Figure 6 polymers-14-01379-f006:**
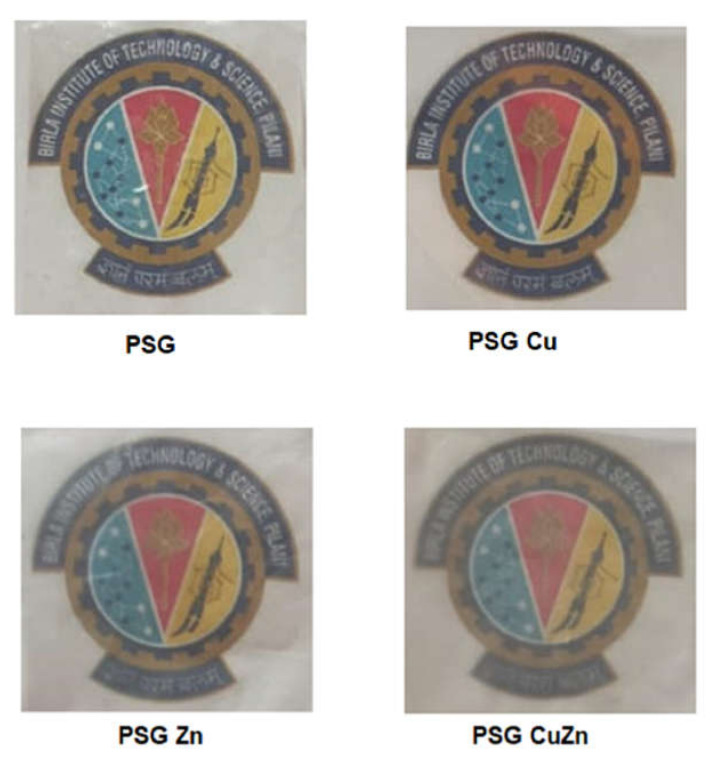
Images displaying the transparency of polymer films to visible light.

**Figure 7 polymers-14-01379-f007:**
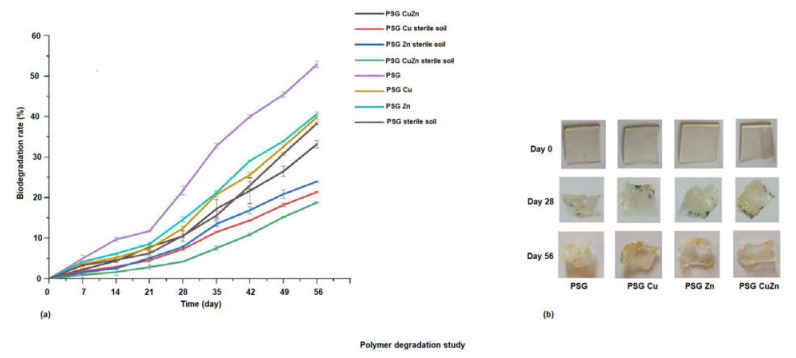
(**a**) Degradation of polymer films incubated in sterile and non-sterile soil over 56 days. (**b**) Polymer films exhibiting visible change on degradation.

**Figure 8 polymers-14-01379-f008:**
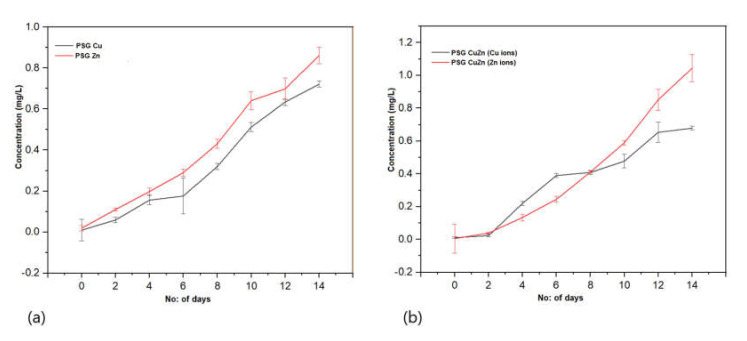
Release profile of Cu^+2^ and Zn^+2^ ions from (**a**) PSG-Cu and PSG-Zn films. (**b**) PSG-CuZn film when kept submerged in water for 14 days. The analysis was conducted using a Perkin Elmer Optima 8300 ICP-OES.

**Figure 9 polymers-14-01379-f009:**
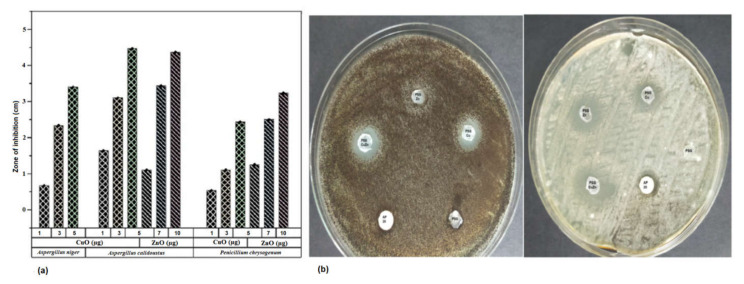
(**a**) ZOI exhibited by of the CuO and ZnO nanoparticles on *Aspergillus niger*, *Aspergillus calidoustus*, and *Penicillium chrysogenum* at the concentration of 1, 3, and 5 µg of CuO nanoparticles and 5, 7, and 10 µg of ZnO nanoparticles. (**b**) Antifungal effect of PSG, PSG–Cu, PSG–Zn, PSG–CuZn, and Amphotericin B (20 µg) on *Aspergillus niger* and *Aspergillus calidoustus*.

**Figure 10 polymers-14-01379-f010:**
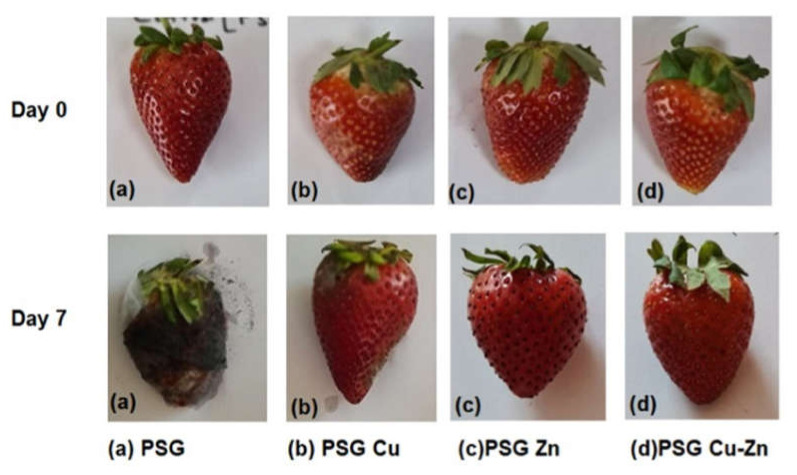
Comparison between the appearance of strawberries wrapped with PSG, PSG–Cu, PSG–Zn, and PSG–CuZn films on day 0 and day 7.

**Figure 11 polymers-14-01379-f011:**
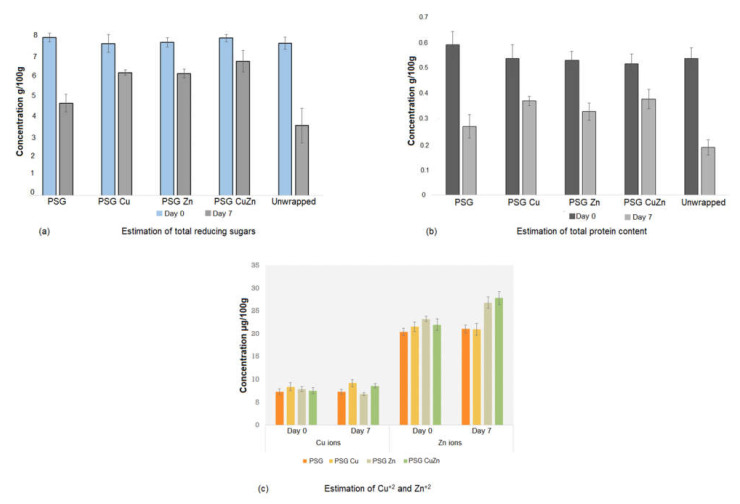
Estimation of the nutritional quality of the food in terms of (**a**) reducing sugars, (**b**) total protein, and (**c**) Cu^+2^ and Zn^+2^ ions in strawberry samples. Anthrone test and Lowry’s method was followed to estimate the total content of reducing sugars and proteins in the sample. The estimate of Cu^+2^ and Zn^+2^ ions was determined using ICP-OES analysis.

**Table 1 polymers-14-01379-t001:** Mechanical properties of the polymer composite films analyzed following the ASTM standard method D-882-88 using a Shimadzu Autograph AGSX-10KN.

Nanocomposite Films	Tensile Strength (MPa)	Young’s Modulus (GPa)	Elongation Break (%)
PSG	18.05 ± 0.038	4.26 ± 0.44	402.08 ± 0.61
PSG–Cu	23.71 ± 0.08	6.51 ± 0.31	324.03 ± 0.21
PSG–Zn	31.83 ± 0.17	7.81 ± 0.068	368.63 ± 0.4
PSG–CuZn	29.12 ± 0.3	8.33 ± 0.34	301.08 ± 0.49

## Data Availability

The data presented in this study are available on request from the corresponding author.

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
