# Peer review of "Metallic Nanoparticle Integrated Ternary Polymer Blend of PVA/Starch/Glycerol: A Promising Antimicrobial Food Packaging Material"

_polymers, 2022, doi:10.3390/polym14071379_

Round 1

Reviewer 1 Report

Comments to authors are listed below:

  1. The abstract lacks to present the significant findings from the results reported in this paper.
  2. The introduction did not present the novelty from this paper clearly, and the applications of this study should include at the end of the introduction section.
    1. The applications and characterisations should be explained clearly by including recent related references. The authors are recommended to include the references below to improve the quality of the introduction:

    https://link.springer.com/article/10.1007/s42452-019-1111-2

    https://www.sciencedirect.com/science/article/pii/S0144861718302571

    https://www.degruyter.com/document/doi/10.1515/ipp-2020-3974/html

  1. The chemical compositions changes are recommended to be investigated by suing FTIR analyser.
  2. The discussion of thermal properties (TGA and DTG thermograms) need to be discussed in details with interpreting the significant numerical values for comparison. They should compare with previous related works to show the improvements in the thermal and mechanical properties.
  3. The conclusions should be supported by the significant results from this study.

Reviewer 2 Report

Dear Authors,

The present study is interesting and aimed at synthesizing a novel biodegradable polymer PVA-starch-glycerol (PSG) incorporated with CuO and ZnO nanoparticles, possessing antimicrobial properties with mechanical and thermal properties comparable to a petroleum-based polymer.
I suggest the authors give more details when discussing the analysis. Can authors highlight CuO / ZnO nanoparticles in SEM images? I suggest that the authors: explain more the process of water absorption; not to discuss the individual characterization analyzes but to make correlations between the results of the performed analyzes.

Reviewer 3 Report

This paper describes the preparation of a CuO and ZnO nanoparticle integrated ternary polymer blend of PVA/Starch/Glycerol and its application as a promising antimicrobial food packaging material. The paper is well written and structured. Also, the experimental results have the associated errors as well as the error bars are included in the plots. However, taking into consideration that the polymer is not novel, because there are many papers about the optimization of the best formulations (see for example: Effect of Glycerol on Thermal and Mechanical Properties of Polyvinyl Alcohol/Starch Blends, Journal of Applied Polymer Science 123(1) 2012.DOI: 10.1002/app.34465), further information about the antimicrobial properties of the polymer and the health risks to humans should be included. The following information should be included:

  • The MIC and MFC and MBC should be obtained and compared with standard antifungal and antibody agents as well with other similar polymers.
  • The health risks of the metals release into food must be assessed and conclusions about the food quality standard regulations must de discussed.
  • Better quality XRD should be shown.
  • Further information (number of samples) about the strawberries used in the assays should be given.

Reviewer 4 Report

This research is dealing with a very hot and interesting topic in coating research. The production of new formulation for food industry is a very attracting field and authors approached it with an innovative research. 

Anyhow, this paper present some issue that must be addressed prior its publication.

Authors must report more data about the reagent used. Which kind of PVA was used? The purity of glycerol? These data MUST be added.

Section 2.3 is not properly formatted.

Line 209: previously described where?

Data reported in figure 3, 8 and 10 must be marked with different letter if data are significantly different from each other. Accordingly, stastitistic test must be run to proper analyze all data set.

A generally remark, author did not put enough care in the assembling of the paper. Data are very promising and they deserve a clearer and appropriate presentation.

Furthermore, a comparison with other systems presented in the literature should be included.

Nevertheless, this paper is very appealing and I warmly endorse its publication after a proper revision.

Round 2

Reviewer 1 Report

No comments

Reviewer 3 Report

The authors have answered satisfactorily to the referees questions.

Reviewer 4 Report

Authors partially fullfilled my request and now the article reached up the minum  grade for publication.